# In Vitro Fermentation Shows Polyphenol and Fiber Blends Have an Additive Beneficial Effect on Gut Microbiota States

**DOI:** 10.3390/nu16081159

**Published:** 2024-04-13

**Authors:** Jordan A. Whitman, Laurel A. Doherty, Ida G. Pantoja-Feliciano de Goodfellow, Kenneth Racicot, Danielle J. Anderson, Katherine Kensil, J. Philip Karl, Glenn R. Gibson, Jason W. Soares

**Affiliations:** 1Soldier Performance Division, U.S. Army Combat Capabilities Development Command (DEVCOM) Soldier Center, Natick, MA 01760, USA; jordan.a.whitman2.civ@army.mil (J.A.W.); laurel.a.doherty.civ@army.mil (L.A.D.); ida.g.pantojafeliciano.civ@army.mil (I.G.P.-F.d.G.); kenneth.racicot.civ@army.mil (K.R.); 2Combat Feeding Division, U.S. Army Combat Capabilities Development Command (DEVCOM) Soldier Center, Natick, MA 01760, USA; danielle.j.anderson.civ@army.mil (D.J.A.); katherine.r.kensil.civ@army.mil (K.K.); 3Military Nutrition Division, U.S. Army Research Institute of Environmental Medicine (USARIEM), Natick, MA 01760, USA; james.p.karl.civ@health.mil; 4Food and Nutritional Sciences, University of Reading, Reading RG6 6AH, UK; g.r.gibson@reading.ac.uk

**Keywords:** prebiotics, gut microbiome, carbohydrate, supplementation, extracts, bacteria, metabolites

## Abstract

Polyphenols and fermentable fibers have shown favorable effects on gut microbiota composition and metabolic function. However, few studies have investigated whether combining multiple fermentable fibers or polyphenols may have additive beneficial effects on gut microbial states. Here, an in vitro fermentation model, seeded with human stool combined from 30 healthy volunteers, was supplemented with blends of polyphenols (PP), dietary fibers (FB), or their combination (PPFB) to determine influence on gut bacteria growth dynamics and select metabolite changes. PP and FB blends independently led to significant increases in the absolute abundance of select beneficial taxa, namely *Ruminococcus bromii*, *Bifidobacterium* spp., *Lactobacillus* spp., and *Dorea* spp. Total short-chain fatty acid concentrations, relative to non-supplemented control (F), increased significantly with PPFB and FB supplementation but not PP. Indole and ammonia concentrations decreased with FB and PPFB supplementation but not PP alone while increased antioxidant capacity was only evident with both PP and PPFB supplementation. These findings demonstrated that, while the independent blends displayed selective positive impacts on gut states, the combination of both blends provided an additive effect. The work outlines the potential of mixed substrate blends to elicit a broader positive influence on gut microbial composition and function to build resiliency toward dysbiosis.

## 1. Introduction

Gut microbiome community composition and function is influenced by dietary intake, with diet–microbiome interactions linked with maintaining gut homeostasis and limiting dysbiosis associated with inflammatory conditions [1]. As such, there is substantial interest in developing dietary strategies that can improve health by favorably modulating the gut microbiota and by promoting the growth of beneficial taxa and production of bacterial-derived metabolites [2]. Of particular interest are dietary fibers and polyphenols, which have gained traction as nutritional supplements that positively impact gut microbial community dynamics [3,4,5]. Large-molecular-weight polyphenols and fermentable fibers are poorly absorbed by the upper digestive tract, reaching the large intestine with minimal degradation by digestive enzymes [6,7].

The majority of previous studies have focused on individual polyphenol classes (e.g., cranberry proanthocyanidins) and fiber constituents (e.g., resistant starch), which have provided significant data on the effects of individual compounds on gut microbial composition, the production of beneficial metabolites, and the reduction of pro-inflammatory metabolites [8,9]. Studies exploring “whole foods” like fruits, vegetables, and whole grains to understand how natural combinations of polyphenols and fibers positively impact gut microbiota and host gut health have also emerged without distinction to the specific fiber and/or polyphenol sources [10]. Focused studies understanding how specific blends of polyphenol and fibers, from varying dietary sources, impact bacteria diversity and metabolite production are limited, and a strategic rationale for the combination of polyphenols and fibers requires additional research to understand the potential for combinatorial effects on gut health. This is of relevance in situations where whole-food consumption is not always practical such as in a military operational setting.

In the current study, we have explored, through in vitro fermentation, the impact of polyphenol (PP) and fiber (FB) blends on gut microbial dynamics, pro/anti-inflammatory metabolite production, and antioxidant capacity. The PP and FB blends were chosen to incorporate compounds with a diversity of structures that, individually, have been shown to beneficially impact gut microbiota composition and function. The goal was to elicit broader favorable effects on gut microbiota community composition and metabolic activity than any individual fiber or polyphenol source without introducing antagonistic effects. The PP blend contained blueberry, cranberry, green tea, and cocoa powders/extracts providing a diverse range of polyphenol structures that have been shown to positively impact gut states, in particular antioxidant capacity [11,12,13,14]. Constituents of the FB blend were chosen to incorporate a diversity of fiber structures that beneficially impact gut microbiota composition and function, particularly through the stimulation of short-chain fatty acids (SCFAs) and beneficial bacterial taxa [15,16]. The FB blend contained high-amylose maize starch and two prebiotic substrates, galacto-oligosaccharides (GOS) and oligofructose-enriched inulin, which provided varied polysaccharide types and lengths. An in vitro model capable of simulating the physiological conditions of the ascending domain of the large intestine [17] was employed to provide a broad experimental capacity. This allowed us to study community dynamics and rates of metabolite production as a function of dietary supplementation in a systematic manner to understand blend-specific effects on gut microbial dynamics [18]. The hypothesis was that the combination of the polyphenol and fiber blends would promote synergistic outcomes for the growth of beneficial taxa, production of beneficial metabolites, and reduction of pro-inflammatory compounds and bacterial taxa associated with dysbiosis.

## 2. Materials and Methods

### 2.1. Study Population and Sample Collection

Fecal samples were sourced from a previous study, collected from 30 healthy adults (habitual diet control group) aged 18–62 years and without obesity (BMI ≤ 30 kg·m^−2^) who were participating in a randomized controlled trial investigating the effects of consuming a diet of military food rations on the gut microbiota (clinicaltrials.gov NCT02423551) [17,19,20]. Participants had not used oral antibiotics or had colonoscopies within three months of study participation, did not have histories of gastrointestinal disease or regular use of medications impacting gastrointestinal function, and were not following a restrictive diet or attempting to lose or gain weight. Additionally, participants were instructed to discontinue use of any probiotic, prebiotic, or other dietary supplements at least two weeks prior to beginning study participation. The study was reviewed and approved by the US Army Research Institute of Environmental Medicine Institutional Review Board, and all participants provided written informed consent prior to participation. Fecal sample collection was adapted from [21] and protocol has been described in detail in Pantoja-Feliciano et al. [22].

### 2.2. Blend Formulations

For supplementation, polyphenols were extracted from four agricultural products; cocoa seed extract (CocoActiv^®^; 45.1% total phenolics dry wt. Cocoa Extract (CE), Naturex, Avignon, France), wild blueberry powder (4.0% total phenolics dry wt. Gallic Acid Equivalents (GAEs), Naturex, Avignon, France), cranberry extract (Cystricran^®^; 57.2% total phenolics dry wt. GAEs, Naturex, Avignon, France), and green tea leaf extract (100% total phenolics dry wt. GAEs, Naturex, Avignon, France) were selected. Three fermentable fiber sources were also selected: Orafti^®^ Synergy1 (93.2% oligofructose-enriched inulin by dry wt., Beneo GmbH, Mannheim, Germany), Bimuno-galactooligosaccharides^®^ (85% GOS (*w*/*w*), Clasado Biosciences, Reading, UK), and Hi-Maize^®^ 260 (59% resistant starch *w*/*w*, Ingredion, Inc., Bridgewater, NJ, USA). Select polyphenols and fiber sources were combined in ratios comparable to that used within the human study referenced above, either in alcohol-solubilized form (polyphenols) or fermentation medium (fibers), to provide a dose of 2 g polyphenol/day and 30 g fiber/day, respectively (Appendix A). Doses were chosen to align with a human study assessing the effects of fermentable fiber and polyphenol supplementation on intestinal barrier function during environmental stress (clinicaltrials.gov identifier: NCT04111263). The supplemented blends were given the designations of PP for the polyphenol blend and FB for the fiber blend. The combination of the two blends was given the designation PPFB.

### 2.3. Study Design and Supplementation Parameters

An overview of the fermentation study design can be found in Figure 1. Parallel batch fermentations (*n* = 3) were designed to include vessels for blend-specific supplementation, non-supplemented (fecal-only, F), and fecal-deficient (medium-only, NF) using an HEL BioXplorer 100 (HEL Group, Borehamwood, UK) 8-vessel parallel bioreactor simulating the ascending colon (pH 5.5). The fermentation medium, Colonic Complex Medium (CCM) was prepared based on Pantoja-Feliciano et al. [22] with slight modifications: addition of resazurin (1 µg/L). The medium was added to the fermentation vessels (125 mL/vessel) with pH, temperature, and oxidation–reduction potential (Applikon Biotechnologies, Foster City, CA, USA) probes attached. Vessels were autoclaved for 35 min at 120 psig and attached to the bioreactor for overnight equilibration under continuous oxygen-free nitrogen (20 psig, 5 mL/min) at 37 °C and with constant agitation (450 rpm). After equilibrium, pH values were corrected for calibration drift and pH control was initiated (pH 5.5 ± 0.1) with bioreactor-controlled adjustments using 1 N NaOH and 0.2 N HCl. Prior to supplementation, CCM volume equivalent to total supplementation volume was removed from each vessel. PP solubilized in methanol was added to the appropriate vessels (5 mL/vessel) using a syringe equipped with an 18-gauge needle and equilibrated overnight. After overnight stabilization, fiber, suspended in CCM, was added to appropriate vessels (9 mL/vessel) and vessels were allowed to stabilize for 30 min. Equal volumes of methanol and CCM were added to non-supplemented vessels corresponding to PP and FB, respectively.

Fecal samples from 30 participants were thawed and pooled in serum bottles at 20% *w*/*v* inside an anaerobic chamber (Coy Labs, Grass Lake, MI, USA) to generate fecal inoculum prior to inoculation. Pooled fecal samples simulate a universal gut microbiome and allow for increased diversity in the microbial population by maximizing species lower in abundance from more subjects [23]. After vessel supplementation with PP, FB, or PPFB, vessels were inoculated with pooled stool through headplate septum using syringes equipped with 18-gauge needles. Fecal-deficient control vessels were inoculated with sterile 0.1 M phosphate buffer pH 7.2 with 15% (*w*/*v*) glycerol. Vessels were sampled at 0, 5, 10, and 24 h, representing community lag (0 h), logarithmic (5 h, 10 h), and stationary (24 h) growth phases, and sample aliquots were stored at −80 °C for further analysis.

### 2.4. Targeted Keystone Bacterial Taxon Analysis

DNA extractions were performed using the QIAMP Power Fecal Pro DNA Extraction Kit (QIAGEN Inc., Germantown, MD, USA). Quantification of DNA (ng/µL) was performed using a Nanodrop One™ instrument (Thermo Fisher Scientific Inc., Waltham, MA, USA). To determine absolute abundance, standard curves were generated using DNA extracted from pure cultures of eight organisms purchased from ATCC (American Type Culture Collection, Manassas, VA, USA): *Bifidobacterium animalis subsp. lactis* 700451, *Lactobacillus reuteri* 23272, *Eubacterium rectale* 33656, *Ruminococcus bromii* 27255, *Akkermansia muciniphila* BAA-835, *Blautia hansenii* 27752, *Dorea* spp. BAA-2280, and *Faecalibacterium prausnitzii* 27768. The specific taxa were selected based on previous studies [17,22]. Briefly, organism-specific quantitative polymerase chain reaction (qPCR) primers were selected for the eight organisms included [17,24] (Appendix A). Serial dilutions (10-fold) were prepared using DNase- and RNase-free water, and 2× Forget-Me-Not qPCR Master Mix (Biotium, Hayward, CA, USA) was added in the reactions. The iCycler iQ Optical module™ version 3.1 (Bio-Rad Laboratories, Hercules, CA, USA) software was used to quantitate each qPCR reaction. For all reactions, target qPCR efficiency was between 80% and 100% [25]. Genome size for each microorganism was used to calculate copy number for each organism [18,26]. Raw results from qPCR were log-transformed to calculate copy number/mL.

### 2.5. Metabolite Analysis

Fermentate concentrations of acetic, butyric, propionic, valeric, isovaleric, and isobutyric acids were analyzed using the approach described by [27,28,29] with slight modifications. Briefly, aliquots were thawed, homogenized, and acidified using 50% sulfuric acid. SCFAs were then extracted using diethyl ether by removing the organic layer after centrifugation. Ethyl butyric acid was added as an internal standard before storing at −80 °C until analysis. SCFAs were quantified using an Agilent 7890A GC system with Flame Ionization Detection (Agilent J&W DB-FFAP column dimensions: 60 m × 250 μm × 0.25 μm, Agilent Technologies, Santa Clara, CA, USA). Calibration standards were included for each fatty acid and used for peak identification and quantification.

### 2.6. Influence of Supplementation on Indole

Indole concentrations were measured in fermentates in triplicate using the Indole Assay Kit, MAK326 (Sigma-Aldrich Co., St. Louis, MO, USA). Color intensities were quantified at 565 nm, room temperature using Biotek Powerwave HT (Agilent Technologies, Santa Clara, CA, USA). Indole concentrations were calculated from the slope of a standard curve quantified by subtracting the plate blank from the standard values and graphing against indole concentrations in slope–intercept form. The fecal-deficient sample readings were subtracted to determine net concentrations of indole in corresponding fecal vessels.

### 2.7. Influence of Supplementation on Ammonia

Ammonia concentrations were measured in fermentates using the non-enzymatic Ammonia Assay kit, ab102509 (Abcam Inc., Waltham, MA, USA). Prior to analysis, all samples were filtered using 10 kD spin columns, ab93349 (Abcam Inc. Waltham, MA, USA) to remove proteins and lower background levels of ammonia. Dilutions were required to bring sample concentrations within the range of the standard curve (0 mM–10 mM). Plates were incubated for 30 min at 37 °C with samples measured in triplicate. Color intensities were quantified at 670 nm using Biotek Powerwave HT (Agilent Technologies, Santa Clara, CA, USA). Interference from reagents present in the ammonium chloride standard was compensated by subtracting standard-deficient wells from all readings. This was important as background readings can be significant in fermentates. Unknown sample values were calculated from the ammonium chloride standard curve slope. The fecal-deficient sample readings were subtracted to determine net concentrations of ammonia in corresponding fecal vessels.

### 2.8. Influence of Supplementation on Antioxidant Capacity

Antioxidant capacity was measured in fermentates using the Ferric Reducing Antioxidant Power (FRAP) assay kit, MAK369, Sigma-Aldrich Co. (St. Louis, MO, USA). Prior to running this assay, an initial extraction step is required, using an acid–methanol solution (prepared as 70:29.5:0.5 mixture of methanol:ultrapure water:1 M HCl). The acidic conditions allow for the dissociation of Fe^3+^ from the protein complex. In this study, plates were incubated at 37 °C for 60 min with samples measured in triplicate. Color intensities were quantified at 594 nm using Biotek Powerwave HT (Agilent Technologies, Santa Clara, CA, USA). Antioxidant capacity was calculated using a standard curve, after subtracting the negative control, and plotting them using the slope–intercept form. The raw values for the unknown samples were used to calculate unknown antioxidant capacity from the standard curve.

### 2.9. Statistical Analysis

All statistical analyses were performed with RStudio (RStudio 2023.3.0.+386; Posit Software PBC, Boston, MA, USA). Changes in metabolite data were analyzed using 2-way repeated-measures ANOVA with treatment groups (F, FB, PP, PPFB), time points (0, 5, 10, and 24 h), and their interaction included as within-subjects fixed factors. In cases where the interaction was statistically significant, between-group comparisons were conducted using 1-way repeated measures ANOVA with Bonferroni corrections. The assumption of normality was verified using the Shapiro–Wilk test. Mauchly’s sphericity test was used to verify equal variances. For targeted bacterial taxon analysis and SCFA production, 1-way ANOVA with Tukey–Kramer HSD was used for pairwise comparisons. Statistical significance was defined as *p* ≤ 0.05. All graphs were generated using JMP^®^ 15.2.0 (466311) (SAS Institute Inc., Cary, NC, USA).

## 3. Results

### 3.1. Effects of Polyphenol and Fiber Supplementation on Bacterial Abundance

Microbial compositional changes in eight targeted beneficial taxa after polyphenol and fiber supplementation were determined quantitatively using qPCR. Across all the selected taxa, supplementation led to differential changes in abundance, as shown in Figure 2. Supplementation with PP and FB blends led to collective differences in *Bifidobacterium* spp. abundance relative to F (time-by-treatment interaction, *p* = 0.047; Figure 2A). Additionally, time-dependent analysis between treatment groups illustrated elevated abundances of *Bifidobacterium* spp. in PP, FB, and PPFB at 5 h and 10 h (*p* < 0.05) compared to F and, for FB and PPFB specifically, at 10 h and 24 h (*p* < 0.05), but a change in abundance was not statistically significant with PP at 24 h (Appendix A). Significant differences were not observed between supplemented vessels. *Lactobacillus* spp. in supplemented vessels also showed notable increases in abundance relative to F (time-by-treatment interaction, *p* = 0.021; Figure 2B) and, more specifically, higher abundances in relation to PP, FB, and PPFB at 10 h and 24 h (*p* < 0.02; Appendix A), with significant differences observed between PP and PPFB at 24 h (*p* = 0.046). Additionally, there was a significant difference between PPFB and F at 5 h (*p* = 0.032).

*Dorea* spp. abundance changes were driven by PP and PPFB supplementation (time-by-treatment interaction, *p* = 0.049; Figure 2C), resulting in sustained abundances in PP and PPFB at 10 h and 24 h (*p* < 0.05) compared to both FB and F. FB supplementation showed similar growth trends to F with decreases in abundance after 5 h. Additionally, differences between PPFB and F were seen at 5 h (*p* = 0.01) (Appendix A). *Ruminococcus bromii* showed increased abundance after FB and PPFB supplementation (time-by-treatment interaction, *p* = 0.002) (Figure 2D). This led to higher abundances of *R. bromii* in FB and PPFB at 10 h (*p* < 0.01) and 24 h (*p* < 0.05) compared to F. Although *R. bromii* abundance was slightly higher at inoculation (T = 0 h) within PP supplementation relative to other treatments, PP showed similar growth trends to F with decreased abundance as the fermentation time approached 24 h. Significant differences were seen between PP and FB at 10 h (*p* = 0.017) and between PP and PPFB at 10 h (*p* = 0.015) (Appendix A).

For the other four taxa, no significant interactions between time and treatment were observed (Appendix A). However, although a two-way interaction was not evident, interestingly, *Faecalibacterium prausnitzii* upon PP treatment trended differentially to the other blends at 24 h while the other three organisms converged (Appendix A).

### 3.2. Effects of Polyphenol and Fiber Supplementation on Metabolite Concentrations

#### 3.2.1. Changes in Total SCFA Concentrations

Mean concentrations of total SCFAs (acetic, propionic, and butyric acids) increased from 0 h to 24 h differently within each treatment group (Figure 3, Appendix A). There was a significant time by treatment interaction (*p* < 0.001). Pairwise comparisons between treatment within each time point showed significant increases in FB and PPFB compared to F at 10 h and 24 h (*p* < 0.05, Appendix A). However, significant statistical differences were not evident between PP and F (*p* > 0.05, Appendix A) across all fermentation time points. Significant differences were not seen between PP and FB compared to PPFB (*p* > 0.05, Appendix A) at any residence time. The ratios of acetic acid, propionic acid, and butyric acid were similar across all blends; however, butyric acid was slightly elevated in FB relative to PP and PPFB at 24 h (Appendix A). At 10 h, butyric acid proportions in PP were 2-fold and 3-fold higher than those seen in PPFB and FB, respectively. Although total SCFA content demonstrated statistical significance in select instances, mean concentrations at individual time points for SCFAs (acetic, propionic, and butyric acids) were not significantly different across all blends as a function of time (*p* ≥ 0.05, Appendix A).

#### 3.2.2. Changes in Indole Concentrations

Indole analysis was performed to measure protein metabolite production. The mean concentration of indole (*n* = 3) decreased within F from 0 h to 10 h and then increased at 24 h (Appendix A). Indole concentrations were high in both FB and PP compared to F and PPFB at 0 h (Figure 4). Supplementation with each of the three blends lead to notable decreases in indole concentrations at 5, 10, and 24 h (Appendix A). There was a significant time by treatment interaction (*p* < 0.001, Figure 4) in indole concentrations. Pairwise comparisons between treatments within each time point showed a significant statistical difference in PP compared to F (*p* = 0.044) at 0 h. At 24 h, there were significant statistical differences between FB (*p* = 0.010), PP (*p* = 0.008), and PPFB (*p* = 0.003) compared to F (Figure 4). Additionally, significant differences between PP and PPFB were seen at 24 h (*p* = 0.002, Figure 4). Pairwise comparisons across time, within treatment, showed significant decreases from 0 h to 24 h in FB (*p* < 0.05) and PPFB (*p* < 0.001, Appendix A). The linear slopes depicting rates in indole concentrations over time (Figure 4, inset) show mean concentrations of indole (µM) decreasing across the 24 h in the three supplemented blends, with similarities between FB and PPFB concentrations, while increasing in non-supplemented samples.

#### 3.2.3. Changes in Ammonia Concentrations

Ammonia concentrations remained consistent within F at 0 h, 5 h, and 10 h, with an increase at 24 h, although not significantly between time points (Figure 5, Appendix A). Supplementation with PP, FB, and PPFB significantly decreased ammonia concentrations after 24 h of fermentation. There was a significant time by treatment interaction (*p* < 0.001) in ammonia concentrations. Significant differences were observed between FB and F (*p* = 0.020) at 24 h. PP supplementation also led to a significant decrease in ammonia production at 24 h compared to F (*p* = 0.032). At 24 h, ammonia concentrations were significantly lower in PPFB compared to F (*p* = 0.016) but without any statistical significance when comparing to FB and PP. Pairwise comparisons across time, within treatment, showed that significant differences in ammonia concentrations were not observed from 0 h to 24 h (Appendix A). Linear slopes (Figure 5, inset) show positive rates of increase in ammonia production over 24 h for F and PP; however, supplementation with FB led to a slower rate of ammonia production while PPFB displayed a more rapid decreasing rate in ammonia production.

#### 3.2.4. Changes in Antioxidant Capacity

Antioxidant capacity remained constant within F and FB from 0 h to 10 h (Figure 6, Appendix A). In PP and PPFB, antioxidant capacity peaked at 5 h and decreased at 10 and 24 h. There was a significant time by treatment interaction (*p* < 0.001) when it came to antioxidant capacity. Pairwise comparisons across treatment show that there were significant differences in PP (*p* < 0.05) and PPFB (*p* < 0.05) compared to FB and F at each time point from 0 h to 24 h (Figure 6). There were no significant differences between FB and F within each time point. There were also no significant differences between PPFB and PP within each time point. The presence of FB in PPFB did not impact antioxidant capacity. Pairwise comparisons across time showed that there were significant differences in PP (*p* = 0.012) and PPFB (*p* = 0.043) between 10 h and 24 h (Appendix A). In F, there was a significant decrease in antioxidant capacity at 24 h compared to 0 h (*p* = 0.042). FB did not contribute to significant changes in antioxidant production across time. No significant differences in antioxidant levels were observed in the fecal-deficient NF samples (Figure 6, inset) across the 24 h fermentation, indicating that antioxidant capacity changes observed with supplementation were due to PP and PPFB blends with fecal microbiota present.

## 4. Discussion

The main aim of this study was to understand the independent and combined effects of supplementing an in vitro model of the large intestine with polyphenol and fiber blends on the growth of beneficial taxa, change in SCFA production, reduction in the concentration of potentially harmful metabolites, and increase in antioxidant capacity.

Growth dynamics of select gut taxa were driven by supplementation with both FB and PP, impacting eight selected taxa differentially. The selected taxa included some keystone and other lower abundant taxa with well-characterized metabolic niches, and these have been shown to assist in metabolizing complex nutrients from the host diet into metabolites for microbial cross-feeding and host utilization [17]. They included RS degraders *Ruminococcus bromii* and *Blautia coccoides-Eubacterium* group; saccharolytic *Dorea* spp., *Lactobacillus* spp. and *Bifidobacterium* spp.; mucin-degrader *Akkermansia muciniphila*; and butyric acid producers *Eubacterium rectale* and *Faecalibacterium prausnitzii.* The taxon changes were seen during the log phases of the fermentation. This was in line with similar in vitro studies comparing the prebiotic effects of individual fibers to a fiber blend [30] with similar changes to select gut bacteria taxa, production of metabolites like SCFAs, and total gas production.

In addition, similar changes were observed when studies employed supplemention their materials with individual polyphenol sources from our PP blend. Fogliano et al. supplemented an in vitro three-stage culture system with cocoa polyphenols, leading to the increased abundance of *Bifidobacterium* spp. and *Lactobacillus* spp. [31]. Both cranberry and blueberry were included in our polyphenol blend due to their high antioxidant content and diverse polyphenol profiles. Similarly, Ntemiri et al. supplemented an in vitro model with polyphenol-rich fractions purified from whole blueberry, leading to significant increases in *Bifidobacterium* spp. and *Feacalibacterium prausnitzii* abundance [32]. Solch-Ottaiano et al. showed that cranberry polyphenols, supplemented in a cross-over study including healthy adults, led to the increased abundance of *Faecalibacterium prausnitzii* in subjects [33]. Zhang et al. showed that supplementing fecal fermentations with catechins found in green tea significantly increased the abundance of *Bifidobacterium* spp., *Lactobacillus*, and *Enterococcus* and increased the production of SCFAs in vitro [34]. These results are consistent with the results seen herein for bacterial taxon changes as a function of PP blend supplementation.

Some studies evaluated polyphenols as part of a whole food, which included fibers naturally present in the food [35]. Our study evaluated the polyphenols and fiber blends both separately and in combination and revealed that the PPFB blend provided an additive positive impact on select taxa when mixing the two blends together in vitro relative to the blends independently. It was important to not immediately assume that the combination of PP and FB would lead to additive effects because there was also a possibility that there could be negative effects associated with combining the two blends due to changes in the competition for resources or potential antimicrobial effects.

The production of beneficial metabolites like short-chain fatty acids (SCFAs) is driven by the fermentation of dietary fiber and proteins by gut bacteria. As the primary metabolite of gut bacterial fermentation, SCFAs play a key role in host homeostasis and regulating bacterial community dynamics [36]. In our work, the supplemented blends lead to more significant changes in total SCFA production rather than changes in individual SCFA production. Similar in vitro studies have observed that total concentrations of SCFAs consistently increased when vessels were supplemented with high concentrations of dietary fiber, driven by fiber source and dosage [37]. Of particular interest is butyric acid, which is produced through saccharolytic fermentation by multiple gut commensals including *Ruminococcus bromii*. Changes in the abundance of these taxa have been shown to increase with fiber supplementation accompanied by a related increase in butyric production. FB supplementation was the key driver of increases in total SCFA production. In our study, the increased production of SCFAs in FB supplementation altered butyric concentrations. This was likely linked to the increased abundance of butyric acid producers such as *Rumminococcus* upon FB supplementation. It is also well cited in literature that the expected SCFA molar ratio of acetic acid, propionic acid, and butyric acid is 60:20:20 [38]. These ratios were observed in all treatment groups across the 24 h fermentation. Previous in vitro studies showed similar results from fiber supplementation. Wang et al. supplemented in vitro batch cultures with a fructan prebiotic, leading to an increase in the concentrations of the SCFAs produced [39].

Studies have also shown that polyphenol supplementation benefits the growth of SCFA-producing bacteria in vivo [40] and in vitro [41]. Here, supplementation with PP led to SCFA concentrations approaching significance at the 10 h and 24 h time points relative to F. The lack of significant changes in SCFA production over time warrants further investigation to determine whether combining the polyphenols had any antagonistic effects on SCFA production. Changes in SCFA production from PPFB were very similar to those seen in FB with elevated total SCFA amounts and comparable ratios, indicating that the primary metabolic impact on the community is centered around the FB components. Havlik et al. saw similar effects of mixing fibers with polyphenols in vitro, affecting the production of phenolic metabolites and SCFAs [42].

Like saccharolytic fermentation, protein fermentation by gut bacteria produces SCFAs; however, these metabolites are accompanied by branch-chained fatty acids (BCFAs) and pro-inflammatory compounds like amines, ammonia, hydrogen sulfide, indole-compounds, and phenols [43]. More specifically, the metabolism of the amino acid tryptophan by gut bacteria generates indole derivatives, which can be beneficial (Indole 3-Propionic Acid) or detrimental (Indoxyl Sulfate) to host health [44]. Supplementation with prebiotics has been shown to decrease the production of pro-inflammatory metabolites from different diets in vitro [39]. The increase in saccharolytic activity by gut bacteria counteracts some of the detrimental effects of a proteolytic environment found in various disease states [45]. Both the results for ammonia and indole herein suggest that FB contributed to the decrease in pro-inflammatory metabolite production. With the additional dietary fiber in FB, a decrease in pro-inflammatory metabolites was observed, most likely due to the lack of amino acids metabolized by proteolytic bacteria [46]. Dietary fiber has been shown to decrease concentrations of ammonia [47] and have an indirect effect on the proliferation of indole-producing bacteria [48].

PP supplementation led to significant decreases in ammonia and indole concentrations compared to F, but less compared to FB supplementation. These findings may be a result of the short residence time within our batch culture. A fed-batch or continuous culture system may augment the effect, which has been shown in vivo. Goto et al. demonstrated that a 6-week supplementation with tea catechins in elderly patients led to significant decreases in fecal ammonia concentrations and other deleterious metabolites [49]. PPFB decreased concentrations of both indole and ammonia similarly to FB in both cases, suggesting that PPFB response is driven by the inclusion of the fiber blend with potential synergistic benefits to decreasing indole concentrations when including the PP blend.

Polyphenols are known for their antioxidant activity while fiber is generally not directly associated with antioxidant production [50]. Antioxidant capacity has a strong link to positive health states by decreasing the presence of reactive oxygen species [51,52]. In this study, FB supplementation did not directly impact the antioxidant capacity. This is supported by other studies regarding an indirect influence as the presence of dietary fiber promotes the growth of beneficial bacteria in the GI tract and the structure of the dietary fibers also has associations to the bioavailability of antioxidant compounds that reach the lower gut [53]. The high antioxidant capacity in both PP and PPFB across all four time points was driven by higher bioavailability to the antioxidant characteristics of PP [54,55]. The decrease in antioxidant capacity at 24 h in PP samples was likely due to the further hydrolysis of PP metabolites. Antioxidant capacity was elevated in the PP- and PPFB-supplemented media and fecal-deficient samples at 0 h, likely due to the PP solubilization generating reactive species during the dissolution process. The retention of antioxidant capacity during active growth upon PP-blend supplementation is a key finding that shows the ability of PPFB to induce positive gut microbiota states not feasible with FB supplementation independently.

The collective impact of PP and FB on select taxa, changes in the total amount of metabolic byproducts, decreased production of pro-inflammatory compounds, and influence on antioxidant capacity all suggest that PPFB outcomes are not driven by PP or FB but both blends concomitantly. Although a synergistic effect was not evident in this study, the additive outcomes of supplementing with both PP and FB suggest that further studies should be considered for microbiome dietary interventions to build toward a healthy, resilient gut microbiome.

## 5. Conclusions

Blends of select polyphenols and fiber substrates were supplemented into in vitro fecal fermentations to determine the impact of the blends independently and collectively. Supplementation with FB led to increases in *Bifidobacterium* spp., *Lactobacillus* spp., and *Ruminococcus bromii* concentrations and, in addition, increases in SCFA production and decreases in indole and ammonia concentrations, but did not impact antioxidant production. Supplementation with PP led to increases in *Bifidobacterium* spp., *Lactobacillus* spp. and *Dorea* spp. accompanied by increases in antioxidant concentrations and decreases in indole and ammonia. In general, significant synergistic or antagonistic effects from combining PP and FB were not evident; however, the contributions of both blends provided a beneficial additive effect that suggests that PPFB creates positive effects to gut microbiome states related to the inclusion of both PP and FB. The work represents a new supplementation approach of employing tailored blends, rather than individual constituents, for microbiome modulation toward healthy gut states to build resiliency towards gut microbial-derived dysbiosis that may be associated with inflammatory conditions.

## 6. Study Limitations

The biggest limitation was the gap between in vitro and in vivo relevance. This study did not include host functionality, commonly simulated using intestinal cell culture models. The incorporation of mucin in the growth medium partially simulates the host, but it does not imitate the effects of an in vivo mucosal environment on the bacterial community. The passive absorption of metabolites by the human colon was not simulated in our model, and the food supply remained continuous unlike in vivo. Additionally, although fecal inoculum was consistent across vessels, the abundance of certain taxa may vary between vessels immediately after inoculation. This phenomenon has been attributed to variability within inoculation and/or downstream processing and may warrant further consideration. Lastly, there is a high variability when comparing gut microbe consortiums between individual subjects that adds another layer of complexity. Here, the individual differences were minimized by pooling feces from 30 subjects for a more diverse microbial population within the fermentation inoculum. Although ideal for in vitro studies, the results may not apply across all individuals.

## Figures and Tables

**Figure 1 nutrients-16-01159-f001:**
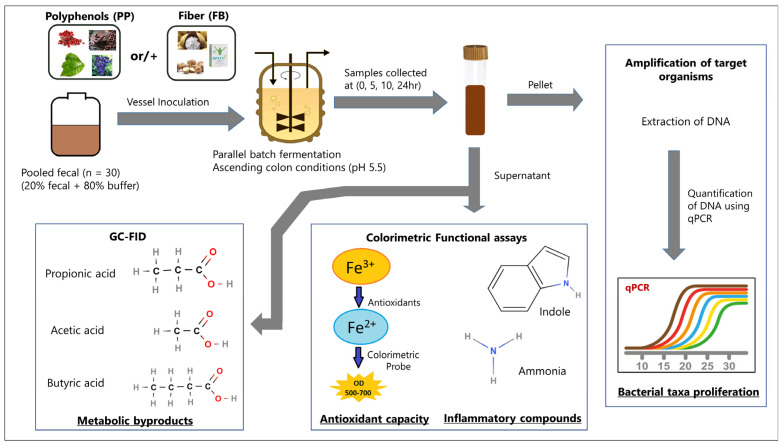
Schematic describing an overview of the experimental workflow.

**Figure 2 nutrients-16-01159-f002:**
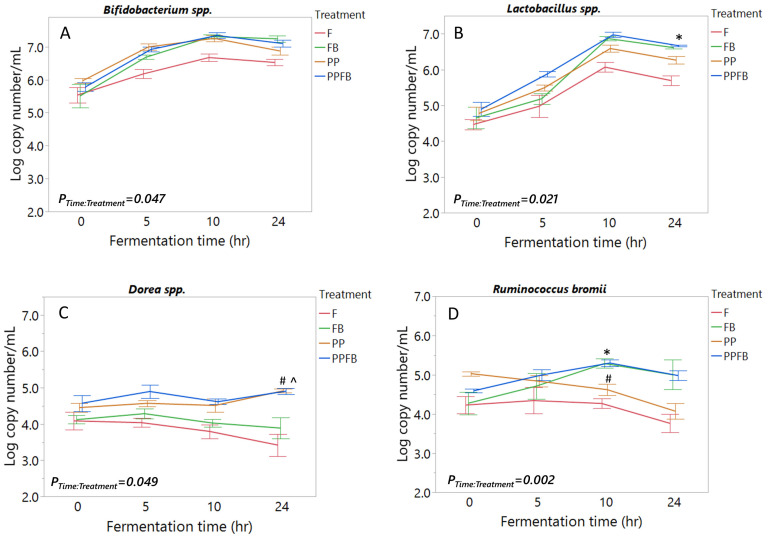
Compositional analysis of selected bacterial taxa and abundance changes over time due to supplementation: *Bifidobacterium* spp. (**A**), *Lactobacillus* spp. (**B**), *Dorea* spp. (**C**), and *Ruminococcus bromii* (**D**). As fermentation residence time increases, differential changes occur within the community. Significant pairwise comparisons by treatment at the same time point (*p* < 0.05): # denotes PP relative to FB (PP-FB); ^ denotes PPFB-FB; * denotes PPFB-PP. Significant difference compared to F at the same time point (*p* < 0.05) can be seen in Appendix A. F = non-supplemented; PP = polyphenol blend; FB = fiber blend, PPFB = PP and FB blend. Data are means ± SEMs (*n* = 3).

**Figure 3 nutrients-16-01159-f003:**
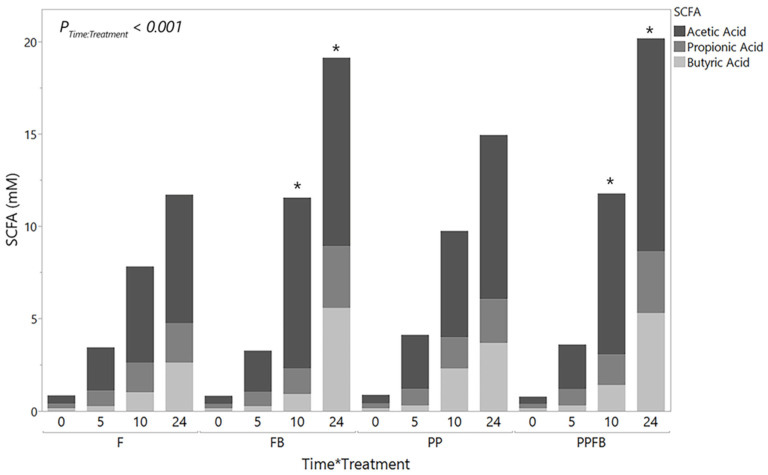
Mean SCFA concentrations over fermentation residence time during supplementation (*n* = 3). As fermentation increases, FB blend shows a marked increase in total SCFA content, similar to PPFB blend. * Significant differences of each treatment relative to F at the same time point (*p* < 0.05). F = non-supplemented; PP = polyphenol blend; FB = fiber blend; PPFB = PP and FB blend.

**Figure 4 nutrients-16-01159-f004:**
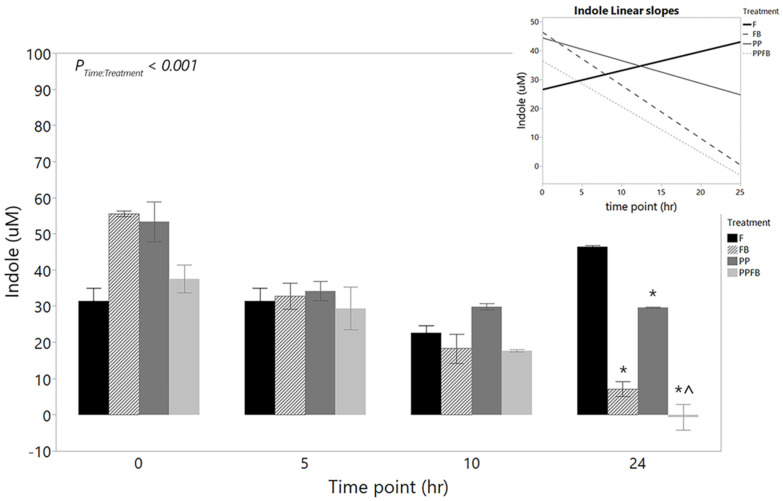
Net changes in indole concentration over time during supplementation including the corresponding linear slope analysis (inset). As fermentation residence time increases, all three blends show a marked decrease in indole, which are statistically different than that of F at 24 h. Linear slopes show differences in indole production rate for the FB and PPFB blends relative to F. * Significant differences across treatment compared to F at the same time point (*p* < 0.05). ^ Significant differences compared to PP at the same time point (*p* < 0.05). Data are means ± SEMs (*n* = 3). F = non-supplemented; PP = polyphenol blend; FB = fiber blend; PPFB = PP and FB blend.

**Figure 5 nutrients-16-01159-f005:**
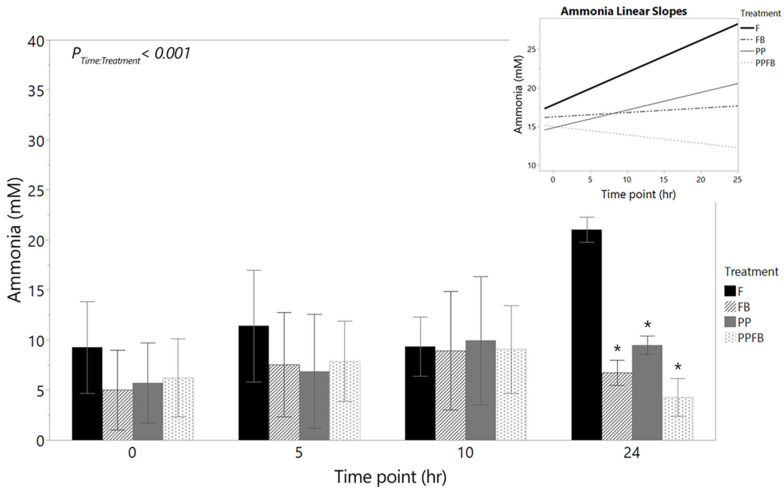
Net changes in ammonia concentration over time during supplementation including the corresponding linear slope analysis (inset). As fermentation residence time increases, FB and PPFB blends show a marked decrease in pro-inflammatory marker ammonia that is statistically different to what is seen in F. Linear slopes show a dramatic rate reduction in ammonia production for PPFB relative to F. * Significant differences across treatment compared to F at same time point (*p* < 0.05). Data are means ± SEMs (*n* = 3). F = non-supplemented; PP = polyphenol blend; FB = fiber blend; PPFB = PP and FB blend.

**Figure 6 nutrients-16-01159-f006:**
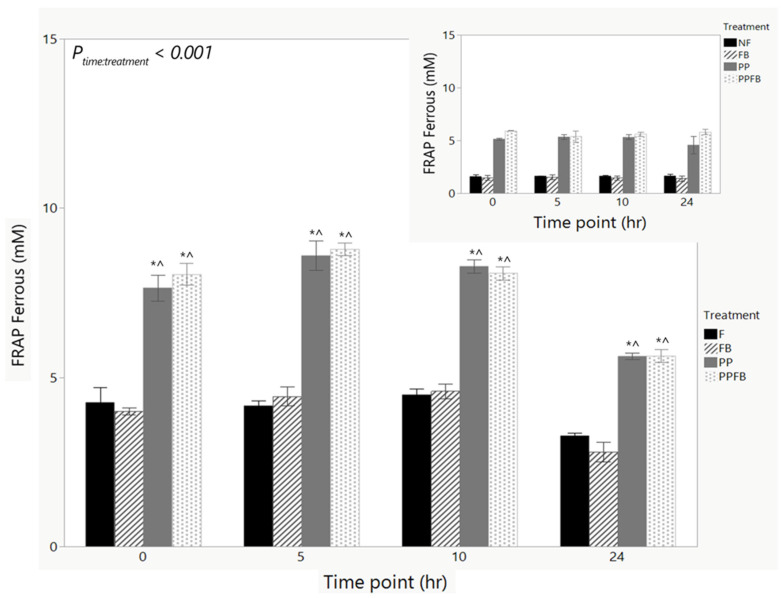
Change in antioxidant capacity over time during supplementation with corresponding capacity in NF samples (inset). As fermentation residence time increases, concentrations of antioxidant marker remain high in PP and PPFB with a marked decrease at 24 h with statistically significant differences to F. FB shows no significant differences to F. NF samples show no changes in antioxidant capacity across the fermentation. * Significant differences across treatment compared to F at the same time point (*p* < 0.05). ^ Significant difference compared to FB at same time point (*p* < 0.05) Data are means ± SEMs (*n* = 3). F = non-supplemented; PP = polyphenol blend; FB = fiber blend; PPFB = PP and FB blend; NF = fecal-deficient.

## Data Availability

All data presented in this study are available on request from the corresponding author. The data are not publicly available due to institutional policies.

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
