# Peer review of "In Vitro Fermentation Shows Polyphenol and Fiber Blends Have an Additive Beneficial Effect on Gut Microbiota States"

_nutrients, 2024, doi:10.3390/nu16081159_

Round 1

Reviewer 1 Report

Comments and Suggestions for Authors

This study explores the effects of polyphenols (PP), dietary fibers (FB), and their combination (PPFB) on microbiota states during in vitro fecal fermentations. The authors discovered that treatments with PP, FB, and PPFB influenced the growth dynamics of gut bacteria. Additionally, FB and PPFB treatments were found to increase short-chain fatty acid (SCFA) concentrations, while PP and PPFB exhibited antioxidant properties. Furthermore, FB, PP, and PPFB treatments reduced indole and ammonia concentrations. These findings suggest that the combined use of these substrates has a synergistic effect, offering new insights into the roles of polyphenol and fiber blends. The following points should be addressed:

1. Insert significant difference markers in Figure 2 to clearly denote statistical differences.

2. Increase the font size in Figures 1 and 2 to improve readability.

3. Clarify the reason for the high log copy number of PP treatment at 0h in Ruminococcus bromii observed in Figure 2.

4. Adjust the Y-axis in Figures 2 and S1 to begin at the minimum log copy number to accurately reflect data distribution.

5. The statement on line 308 regarding ammonia concentrations increasing within F at 0hr, 5hr, and 24hr appears to be inconsistent with the results shown in Figure 5. This discrepancy should be addressed.

6. The Institutional Review Board Statement and Informed Consent Statement on lines 514-515 should either be removed, or, if this item is necessary for the paper, the contents written in the Materials and Methods should also be included here.

Author Response

Authors thank the reviewer for the insightful comments and have provided a point-by-point response in the attached file.

Reviewer 2 Report

Comments and Suggestions for Authors

dear authors,

Your manuscript is very interesting and in line with the latest scientific findings. The microbiota is becoming more and more important.

I Have some  suggestions to you:

MINOR

You must improve resolution of figure 1 and 2;

You can to create a abbreviations list;

You can to write "in vitro" in italic manner.

Mayor

It's not clear why your study could have benefits for human health: For what pathologies? for prevention? For which age groups?

Author Response

(The authors gave the same response as above.)

Round 2

Reviewer 1 Report

Comments and Suggestions for Authors

The authors have satisfactorily addressed the points which I noted.